# The Utility of Explainable AI in Ad Hoc Human-Machine Teaming

**Rohan Paleja**[1], **Muyleng Ghuy**[1], **Nadun R. Arachchige**[1], **Reed Jensen**[2], **Matthew Gombolay**[1]
[1]Georgia Institute of Technology, [2]MIT Lincoln Laboratory
[1]Atlanta, GA 30332, [2]Lexington, MA 02420
{rpaleja3, mghuy3, nkra3}@gatech.edu, rjensen@ll.mit.edu mgombolay3@gatech.edu

## Abstract

Recent advances in machine learning have led to growing interest in Explainable AI (xAI) to enable humans to gain insight into the decision-making of machine learning models. Despite this recent interest, the utility of xAI techniques has not yet been characterized in human-machine teaming. Importantly, xAI offers the promise of enhancing team situational awareness (SA) and shared mental model development, which are the key characteristics of effective human-machine teams. Rapidly developing such mental models is especially critical in ad hoc human-machine teaming, where agents do not have a priori knowledge of others' decision-making strategies. In this paper, we present two novel human-subject experiments quantifying the benefits of deploying xAI techniques within a human-machine teaming scenario. First, we show that xAI techniques can support SA ($p < 0.05$). Second, we examine how different SA levels induced via a collaborative AI policy abstraction affect ad hoc human-machine teaming performance. Importantly, we find that the benefits of xAI are not universal, as there is a strong dependence on the composition of the human-machine team. Novices benefit from xAI providing increased SA ($p < 0.05$) but are susceptible to cognitive overhead ($p < 0.05$). On the other hand, expert performance degrades with the addition of xAI-based support ($p < 0.05$), indicating that the cost of paying attention to the xAI outweighs the benefits obtained from being provided additional information to enhance SA. Our results demonstrate that researchers must deliberately design and deploy the right xAI techniques in the right scenario by carefully considering human-machine team composition and how the xAI method augments SA.

## 1 Introduction

Collaborative robots (i.e., "cobots") and machine learning-based virtual agents are increasingly entering the human workspace with the aim of increasing productivity, enhancing safety, and improving the quality of our lives [16, 22]. In the envisage of ubiquitous cobots, these agents will dynamically interact with a wide variety of people in dynamic and novel contexts. Ad hoc teaming characterizes this type of scenario, where multiple unacquainted agents (in this case, humans and cobots) with varying capabilities must collectively collaborate to accomplish a shared goal [33, 52]. Ad hoc teaming presents a significant challenge in that agents are unaware of the capabilities and behaviors of other agents, and lack the opportunity to develop a team identity, shared mental models, and trust [9, 10, 57]. For example, the human may not be aware of the cobot's possible actions

DISTRIBUTION STATEMENT A. Approved for public release. Distribution is unlimited. This material is based upon work supported by the United States Air Force under Air Force Contract No. FA8702-15-D-0001. Any opinions, findings, conclusions or recommendations expressed in this material are those of the author(s) and do not necessarily reflect the views of the United States Air Force.

35th Conference on Neural Information Processing Systems (NeurIPS 2021).

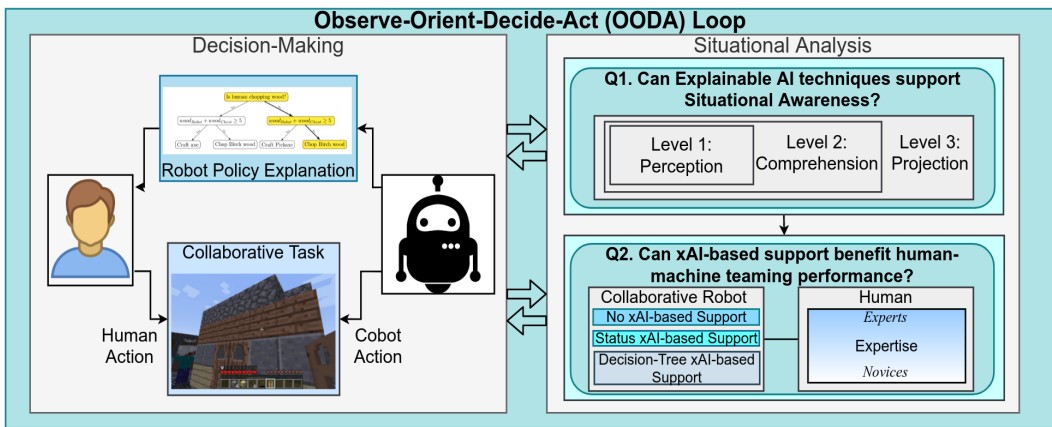

Figure 1: This figure displays an overview of our experimentation in relation to the Observe-Orient-Decide-Act (OODA) loop. On the left, we display the human-machine teaming interaction with both agents taking actions and the cobot outputting a policy explanation to the human teammate. On the right-hand side, we display the two questions assessed by our human-subjects experiments.

and proclivities towards specific activities, limiting her ability to coordinate and plan effectively [35, 48]. Furthermore, this lack of understanding negatively impacts the user's ability to perform situational analysis, impeding a key component of the Observe-Orient-Decide-Act (OODA) loop [7]. For a human teammate to maintain situational awareness (SA) and effectively make decisions in human-machine teaming, the human must maintain an internal model of the cobot's behavior. However, to develop such an understanding of the cobot, the human may have to constantly observe or monitor the cobot's behavior, a costly and tedious process.

Effective collaboration in teaming arises from the ability for team members to coordinate their actions by *understanding* both the capabilities and decision-making criterion of their teammates [37, 47]. For human-human teams, Sebanz et al. [45] states that successful joint coordination among agents depends on the abilities to share representations, to predict other agents' actions, and to integrate the effects of these action predictions. Similarly, we believe these findings should correlate in human-machine teaming. Explainable AI (xAI) techniques, utilizing abstractions or explanations that provide the user insight into the AI's rationale, strengths and weaknesses, and expected behavior [21], can supply the human teammate a representation of the cobot's behavior policy and may assist in the human teammate's ability to predict and develop a collaboration plan. Furthermore, effective human-machine teaming requires the ability for team members to develop a shared mental model (i.e., team members share common expectations about the team coordination strategy, the outcomes of individual strategies, and the individual roles in achieving the team's objective [34]) [35]. xAI techniques offer the promise of enhancing team situational awareness, shared mental model development, and human-machine teaming performance.

Recent work in the machine learning community on xAI has emphasized the importance of interpretability, and post-hoc explainability in enhancing the prevalence of machine learning-based approaches [43]. Other work has even explored simulatability [40, 49], assessing a human's ability to observe a model (e.g., a decision tree, which lends itself to interpretability [14]) and be able to produce the correct output given an input feature. Augmenting machine learning-based systems with some form of intepretability or simulatability can enable these systems to gain human trust [3, 25, 38], an essential quality in high-performance teaming [28]. While prior work has provided approaches for explaining machine behavior through natural language [42], interpretable decision trees [40], and attention-based focusing[55], the utility of collaborative agents augmented with explainable AI techniques in *human-machine teaming* has not been explored.

In this work, we present two novel human-subjects studies to quantify the utility of xAI in human-machine teaming. We assess the ability for human teammates to gain improved SA through the augmentation of xAI techniques and quantify the subjective and objective impact of xAI-supported SA on human-machine team fluency. We first assess if xAI can support the different levels of SA [15] by assessing how different abstractions of the AI's policy support SA within a human-machine

teaming scenario. Second, we study the effect of augmenting cobots with online xAI and assess how different abstractions of the AI's policy affect *ad hoc human-machine teaming performance*. In Figure 1, we present an overview of our experimentation in relation to the Observe-Orient-Decide-Act (OODA) loop. We provide the following contributions:

1. We design and conduct a study relating different abstractions of the cobot's policy to their induced situational awareness levels, measuring how different explanations can help a human perceive the current environment (Level 1), comprehend the AI's decision-making model (Level 2), and project into the future to develop a collaboration plan (Level 3). Our results show that xAI techniques can support situational awareness ($p < 0.05$).

2. We design and conduct an ad hoc human-machine teaming study assessing how online xAI-based support, generated via cobot abstractions, and the human's ability to process higher levels of information affect teaming performance. We find novices benefit from xAI-based support ($p < 0.05$) but are susceptible to information overload from more involved xAI abstractions ($p < 0.05$). Expert performance, on the other hand, degrades with the addition of xAI-based support ($p < 0.05$), indicating that the cost of paying attention to the explanation outweighs the benefits obtained from generating an accurate mental model of the cobot's behavior.

## 2  Related Work

In this section, we discuss relevant prior literature in human-machine teaming and explainable AI.

**Human-Machine Teaming –** The field of human-machine teaming is concerned with understanding, designing, and evaluating machines for use by or with humans [11]. Prior work has emphasized the importance of a shared mental model, noting the benefits of maintaining understanding and predictability within a human-machine team [17, 18, 36]. While these methods have been successful in helping humans identify robot intent, these approaches require modifying the action primitives to make the behaviors more expressive, which can degrade performance, or prior cross-training to enhance understanding, which limits the deployability of cobots.

**Explainable AI (xAI) –** Explainable AI (xAI) is concerned with understanding and interpreting the behavior of AI systems [30]. Prior work has explored utilizing a clear visualization of a policy to help a human form an accurate representation of its capabilities [39], extracting meaning from deep networks through explanations of each layer [38], or generating explanations through separate networks [3, 25]. While the varying types of explanations have been shown to be helpful in limited classification tasks, the utility of these explanations has not assessed in multiagent sequential decision-making problems, such as long-term interactions with cobots.

**xAI in Human-Machine Teaming –** Explainable AI in human-machine teaming is a promising direction as automation with the ability to explain will allow users to better understand the behavior of their AI teammates. Prior work attempts to induce transparency in a human-robot team by the explanation of failure modes [41, 53], synthesis of policy descriptions [24, 54], and the verbalization of experiences [10, 42]. While these approaches are successful are instilling a sense of understanding of robot behavior within the human teammate, the level of collaboration is limited, and these works do not assess the preoccupation cost of online explanations in human-machine teaming. More recently, Bansal et al. [5], Bussone et al. [8], González et al. [20], Lai and Tan [29], Zhang et al. [59], and Zhang et al. [58] investigate the type and accuracy of an xAI explanation to a human's trust and reliance on the setting of human-AI teaming. However, while this prior work deploys xAI-based support in classification problems (utilizing the AI as a recommender system), our task and scenario are widely different in that we consider a complex sequential decision-making and planning problem where the AI is a collaborative agent that actively shapes the world. Anderson et al. [2] provides insight into how different xAI explanations relate to a human player's mental model generation in a simple real-time strategy (RTS) game. Our experiment differs from Anderson et al. [2] in that the human actively teams with the cobot while receiving xAI-based support (Section 5), requiring users to build mental models on the fly. In Anderson et al. [2], the human only acts as an observer, receiving explanations while the AI solely makes decisions within a RTS game.

In our work, we present a foundational set of studies that (1) assess whether xAI-based techniques can support situational awareness in human-machine teaming and (2) quantify the costs and benefits of deploying online xAI techniques in an ad hoc human-machine sequential decision-making problem.

We explore two types of xAI-based support, a short, text-based explanation of the cobot's policy outputs and a decision-tree representation of the cobot's policy. We note that providing human teammates with the complete cobot policy in the form of an interpretable decision tree, as we do in our work, is a type of xAI not investigated in the referenced prior work.

# 3   Human-Machine Teaming Domain

For our experiments investigating the deployment of xAI techniques in human-machine teaming, we utilize the Microsoft Malmo Minecraft AI Project [27]. Minecraft is an open-world environment where players can build structures, craft tools, and play alongside other individuals. Crafting within Minecraft can be classified as a hierarchical task that requires obtaining base materials before generating a more complex object/tool and may require traveling to a crafting table. Building structures is also a hierarchical task as lower layers must be (partially) constructed before the upper layers of a structure can be built. Collaboration among other individuals is highly complex as high-performance teaming requires intent recognition, task coordination, resource/tool sharing, among other personal factors such as game proficiency and trust among teammates.

We generate a $61 \times 61$ grid utilizing the Python API provided by Johnson et al. [27]. Within this grid is two agents, the human and the cobot. The human plays with standard Minecraft controls, utilizing both a keyboard and mouse to continuously move and change her field of view. This teaming scenario is notably different from prior human-machine teaming experiments [9] that restrict human motion to discrete cardinal movements. The cobot is also programmed in continuous space with several action primitives further described in the supplementary material. Planning within this domain is challenging as agents take macro-actions that can take an arbitrary number of time-steps to execute which induces asynchronicity, closely resembling the complexities associated with that of solving a Decentralized Partially Observable Semi-Markov Decision Process (Dec-POSMDP) [1].

Within this domain, the human is unable to explicitly communicate with the cobot (e.g., through chat), and the cobot is able to communicate with the human through a policy abstraction depending on the experiment factor (cf. Section 5). The human also has access to a first-person view of the cobot's screen, providing partial knowledge of the cobot's current location, cobot's current resources, cobot's current action. The cobot has complete access to the human's global state, including current resources, location, and approximate action. The cobot and human are able to share resources through a chest located at a static position, with both agents able to store and take resources depending on need. We provide additional environment dynamics within the supplementary material.

As shown in Figure 2, the human will see up to four elements on her screen. On the top-left, the participant will play Minecraft. On the top-right, the participant has a display showing the first-person view of the cobot. On the bottom of the screen, we utilize the Pygame interface to display information regarding the cobot's policy. On the bottom-left, the participant has a display that may show information regarding the cobot's inference of her action (i.e., the cobot's inference of the human's behavior). On the bottom-right, the participant has a display that may show information regarding the robot's policy (i.e., how the cobot chooses an action).

## 3.1   Human-Machine Collaborative Task

The human and cobot are assigned to build a house with certain specifications. The 4-level house contains eight unique objects including two types of planks, two types of stone, doors, stairs, a fence, and fence gates totaling 89 objects. The latter four items are crafted materials that require combining base materials. Each agent has unique capabilities. The cobot has the ability to collect resources and craft certain items. The human is made aware of all possible behaviors of the cobot. However, the cobot cannot help the human build. The human player is able to collect resources *and* is in charge of building the house. The human is told a priori that her goal is to "Collaborate with the AI as best as possible to complete the house as quickly as possible." However, the human is not strictly specified to build or collect resources in any order.

**Cobot Policy –** The cobot maintains a hierarchical policy. A low-level policy determines the cobot's action, and a high-level policy determines the cobot's inference of the human's action (which the low-level policy is conditioned upon). Both components of the hierarchical policy are decision tree-based policies of depth two and with four leaf nodes. The cobot's high-level policy stays fixed

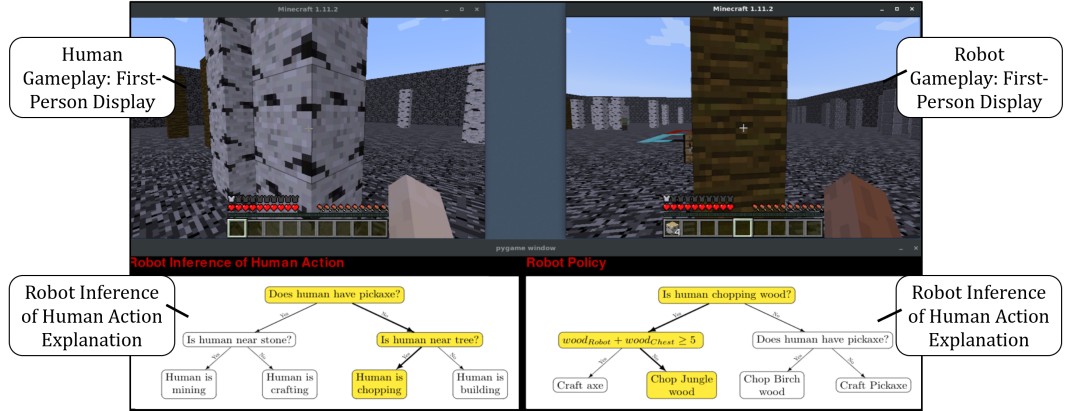

Figure 2: This figure displays a sample gameplay image where the cobot is augmented with the decision-tree explanation. Note this shows IV1:SA1-2-3 condition and IV2:Display Cobot Inference of Human Policy and Cobot Policy in Section 5.

throughout gameplay and is displayed in the bottom-left of Figure 2. The cobot's low-level policy changes throughout gameplay across five phases of play, correlating with different periods of the task (i.e., building different portions of the house). A depiction of one low-level policy in displayed on the bottom-right in Figure 2.

We conducted an initial IRB-approved pilot study with 30 participants to fine-tune the thresholds within the cobot's policy to enforce that the cobot is highly collaborative. We note that while the cobot policy models are hand-designed, the decision-tree based policies serve as a surrogate for the interpretable models produced by interpretable machine learning-based approaches [40, 50].

# 4 Study 1: Relationship Between Explanations and Situational Awareness

Here, we present the details of our first human-subjects study relating different xAI techniques to the three situational awareness levels specified by [15]. We explore the question:

- **Q1: Can xAI techniques be used to support situational awareness (SA)?** If so, which levels of SA are enhanced by xAI?

We review the different levels of situational awareness, study conditions, design, and procedures, and describe the measures employed.

## 4.1 Situational Awareness

Situation awareness (SA) represents a user's internal model of the world around her at any point in time [15], representing the user's ability to perceive the elements of the environment, comprehend their meaning, and project their status in the near future. Situational awareness plays a significant role in mental model alignment among teams and can largely impact performance [46]. Following [15, 44], the three levels of situational awareness induced through xAI techniques are as follows:

- **Level 1: Perception** - Explanation of the current state of the world.
- **Level 2: Comprehension** - Explanation of the agent's current decision-making.
- **Level 3: Projection** - Explanation enabling user to predict future behavior.

The degree to which SA is maintained is typically assessed through the Situation Awareness Global Assessment Technique (SAGAT). The SAGAT prompts a user at a random point within a simulation with a series of fact-based questions to determine her knowledge while blanking out the simulation display.

**Situational Awareness Assessment –** We conduct a SAGAT questionnaire following the recommendations of [44, 51] that provide considerations for adapting the SAGAT towards evaluating situational

awareness in human-machine teaming. Questions for Level 1,2 and 3 consisted of those that determine the current state and action of the human and the cobot, determine the human's understanding of the cobot's current capabilities and features considered in the cobot's decision-making policy, and assess the user's ability to predict the cobot's next action, cobot action given modified inputs, and input requirements to produce a favorable action. Similar to Endsley [15], we maintain a sample of questions for each level of situational awareness. We provide the user three questions per situational awareness level, restricting each question to multiple choice with 2-4 answer choices. We provide the complete list of questions for each SA level within the supplementary material.

While the SAGAT can provide insight into a user's SA, this test is highly intrusive as it requires task interruption. These interruptions can be distracting, interfering with performance [32], increasing stress [31], and likelihood of failure [56]. Accordingly, we choose to assess user SA and the effect of SA separately to reduce intrusiveness in the team fluency study and contribute a novel, targeted SA study across xAI techniques.

### 4.2 Experiment Conditions and Procedures

In this study, we seek to determine how different abstractions of the AI's policy induce different levels of situational awareness. As such, we utilize a $1 \times 3$ within-subjects design varying across three abstractions: 1) No explanation of the robot's hierarchical policy, 2) A status explanation of the cobot's hierarchical policy, and 3) A decision-tree explanation of cobot's hierarchical policy.

- **No Explanation –** In this condition, the human is given no information about the cobot's policy. We note that even within the no explanation condition, the human still has access to the first-person displays of both the robot and human.

- **Status Explanation –** In the status explanation, the human is given information about the cobot's policy relating the cobot's hierarchical policy outputs. The information is given in the form of a short phrase representing the information within the leaf node of the decision-tree explanation.

- **Decision-Tree Explanation –** In the decision-tree explanation, the human is given a decision tree representing the cobot's policy with active edges and decision nodes highlighted. The decision trees displays complete information about the state features that the cobot uses to make decisions and the choice of action. We display the sample game display of this condition in Figure 2.

The experiment was conducted through an online platform where the first page provided the participants with introductory information regarding the human-machine teaming task. Users will then conduct three episodes in which each episode corresponds to a different factor. Each episode consists of $\approx 10$ minutes of gameplay that users must view interrupted by the SAGAT at two-minute intervals (five trials within each episode). The factor/episode ordering were randomized to mitigate confounds.

### 4.3 Results

We recruited 48 participants in an IRB-approved experiment, whose ages range from 18 to 58 (Mean age: 21.69; Standard deviation: 5.61; 18.75% Female). We analyze our data using a repeated-measures ANOVA for the omnibus significance and a Tukey HSD for pairwise comparisons, which includes Bonferroni corrections. Assumptions of normality and homoscedasticity are tested with Shapiro-Wilke and Levene's test respectively. We found that only our Level 2 data failed to meet these parametric assumptions, and thus a Friedman's test with Nemenyi post-hoc (with corrections) was employed instead. We display our findings in Figure 3a. We provide additional analysis details within the supplementary material.

**Q1:** Overall, we find that xAI techniques support higher levels of situational awareness. No significant difference was found in the ability for users to perceive their environment (Level 1) with and without xAI-based support. This indicates that access to a first-person view of teammate gameplay and one's own gameplay is sufficient for perception of the environment. For Level 2 SA, a Friedman's test found significance ($\chi^2(2) = 34.2; p < 0.001$) with the pairwise results showing both status xAI-based support ($p < 0.05$) and decision-tree xAI-based support ($p < 0.001$) significantly increases the ability for the user to comprehend her environment compared to cobots without xAI-based support. Lastly, we also observe an omnibus significance in the Level 3 ($F(2, 94) = 4.01; p < 0.05$), and find that the decision-tree xAI-based support enhances the user's ability to project cobot's behavior into

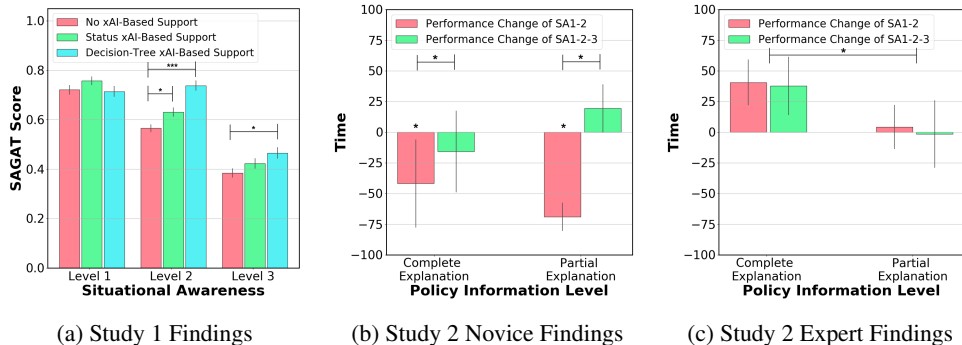

(a) Study 1 Findings  (b) Study 2 Novice Findings  (c) Study 2 Expert Findings

Figure 3: This figure represents the findings of Study 1 (a) and Study 2 (b-c). Figure 3a displays the SAGAT scores across SA levels and xAI abstractions. Figure 3b and 3c display the performance residuals (inverse scale: lower is better) with xAI-based support across policy information levels with respect to the no-explanation condition for novices (Figure 3b) and experts (Figure 3c).

the near future ($p < 0.05$). These results support the hypothesis that xAI techniques enhance SA in human-machine teaming.

# 5 Study 2: Situational Awareness in Ad Hoc Human-Machine Teaming

In this section, we present the details of our second human-subjects experiment exploring:

- **Q2: How does xAI-based support affect ad hoc human-machine teaming performance?**

- **Q3: How does the amount of information regarding the cobot's policy affect ad hoc human-machine teaming performance?**

In Study 2, a participant and a cobot are asked to complete the human-machine collaborative task defined in Section 3.1. We set an exclusion criteria to allow only those with a minimum of 20 hour of lifetime experience in Minecraft to participate. The exclusion criteria removes participants who are unfamiliar with the gameplay controls and ensures that participants are not actively learning to play Minecraft while attempting to collaborate with a cobot they are unfamiliar with. We note that the collaboration between the human and cobot is highly unstructured, and the human's behavior can range from choosing to ignore the cobot to actively perturbing the cobot to understand its behavior more clearly. Below, we review the experimental conditions, design, procedure, and describe the measures employed.

## 5.1 Experiment Conditions

We conduct a 2×3 mixed between-/within-subject design with two factors: **IV1:** Induced SA (Section 5.1) and **IV2:** Policy Information (Section 5.1).

**IV1: Induced Situational Awareness Levels –** The situational awareness levels are aligned via our first study detailed in Section 4. We vary across 1) **SA1: No Explanation**, in which the human is able to perceive the environment, 2) **SA1-2: Status Explanation**, in which the human is able to perceive and comprehend the environment, and 3) **SA1-2-3: Decision Tree Explanation**, in which the human is able to perceive, comprehend, and project the environment.

**IV2: Policy Information Level –** Here, we describe the policy information levels that vary between subjects. Specifically, complete information provides users assistance with Theory of Mind (ToM) [19], reasoning to strategically reason about one's actions in the context of the cobot's decision-making. However, the additional information provided for supporting ToM reasoning may come at the cost of increased complexity and risk of information overload.

- **Condition 1: Display Cobot Inference of Human Policy and Cobot Policy –** The cobot displays both its low-level and high-level policy, providing the user with increased information regarding its decision-making strategy.

- **Condition 2: Display Cobot Policy –** The cobot displays only its low-level cobot policy. This condition provides partial information of how the cobot makes decisions, leaving out the cobot's inference of the human policy.

## 5.2 Procedure

The participant is first briefed on the objective of the experiment, to investigate the impact of online explanations in human-machine teaming. The participant is told that she will be playing with three *different* autonomous agents (use of deception). The participant starts with a pre-experiment survey collecting demographic information, gaming background (experience with video games and Minecraft), and the Big Five Personality Questionnaire [13]. Afterwards, the participant is handed a instructional document regarding specifications for the house and some notes regarding the cobot's behavior. Next, the participant conducts a brief hand-crafted tutorial in Minecraft. Once completed, the participant is tasked with individually building the entire house. The individual house build helps the human gain familiarity with the house specifications, which benefits in the user's task understanding. Once completed, the primary experimentation begins. Users will conduct three episodes in which she will play with three cobots, each of which are programmed the same but with varying xAI-based support and a randomly selected level of policy information. The ordering across xAI-based support levels is randomized to mitigate potential experimental confounds, such as learning effect, fatigue, etc. In each episode, the participant will first be given a sample $\approx$30-second video that describes the upcoming visualization (Figure 2). Following the video, the participant is given a written tutorial describing the xAI-based support. Once completed, the participant will build the specified house in Minecraft with the cobot, a timed task. To conclude the episode, we administer several post-study scales to support our quantitative findings including the Human-Robot Collaborative Fluency Assessment [26], Inclusion of Other in the Self scale [4], Godspeed Questionnaire [6], NASA-TLX Workload Survey [23]. Each scale has been verified for validity in prior work and is used to assess the quality of the human-machine teaming interaction. The Human-Robot Collaborative Fluency Assessment [26] measures team fluency, cobot contribution, trust, positive teammate traits, and perceived improvement through several sub-scales. The Inclusion of Other in the Self [4] scale measures the perceived closeness between teammates. The Godspeed Questionnaire [6] is used to measure perceived likability and perceived intelligence. The NASA-TLX [23] measures the task workload. We provide additional details regarding our procedure in the supplementary material.

## 5.3 Results

We recruited 30 participants under an IRB-approved protocol, whose ages range 18 to 23 (Mean age: 19.97; Standard deviation: 1.30; 16.67% female), with participants randomly assigned to each of the factor levels of **IV2** (the between-subjects variable) with 15 subjects per level. We evaluate participant performance by using the time taken for the human-machine team to finish building the house. Our data was modeled as a mixed-effects ANOVA to capture any possible relation between independent variables across **IV1** and **IV2**. We test for normality and homoschedascity in our data, and employed a corresponding non-parametric test if the data failed to meet these assumptions. We provide details regarding our analysis within the supplementary. We display our objective findings in Figure 3b and Figure 3c.

**Q2:** We conduct a preliminary meta-analysis on our participant data and find two distinctive clusters in participants' individual build times (the separate, pre-test calibration task), implying two distinct categories of participants that vary in gameplay speed. We dichotomize our subject data with an additional categorical variable within our analysis representing each cluster, the first referring to "experts" (i.e., those with higher proficiency in the game of Minecraft) and "novices" (i.e., those with lower proficiency in the game of Minecraft). We find that there are 17 experts and 13 novices within this dataset. While 13 are characterized as novice, it should be noted that our exclusion criteria serves to ensure that all participants are familiar with the gameplay controls of Minecraft and should thus be able to accomplish all key components of the teaming task. With respect to our separate calibration task, we find a significant effect between a participant's teaming ability with the cobot and the participant's individual build speed ($F(1, 26) = 23.5; p < 0.001$).

Novice performance significantly differs across levels of xAI-based support ($F(2, 22) = 4.67; p < 0.05$). We find that novices working with cobots augmented with status xAI-based support (*SA1-2*) are able to complete the human-machine teaming task significantly quicker than users that did not

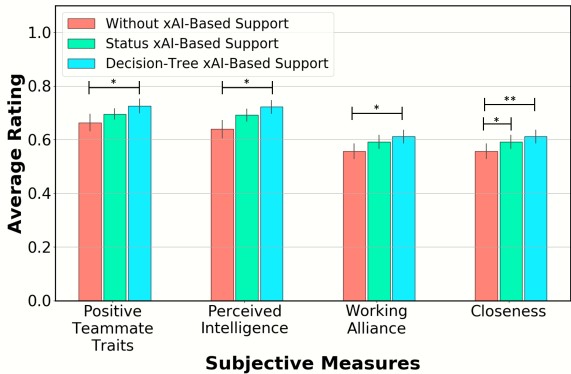

Figure 4: This figure represents the normalized subjective findings of Study 2. We see that all users find cobots with decision-tree xAI-based support to maintain more positive teammate traits, maintain a better working alliance, and are perceived as more intelligent than cobots without xAI-based support. Users also perceive both cobots with status xAI-based support and those with decision-tree xAI-based support as more close than cobots without xAI-based support.

have xAI-based support (i.e., the baseline condition) ($p < 0.05$). While we see that xAI-based support in the form of a short phrase is beneficial, more cognitively intense xAI-based support (decision-tree based support *SA1-2-3*) does not provide novices with benefits in performance. Status xAI-based support *SA1-2* led to significantly improved performance compared to decision-tree xAI-based support (*SA1-2-3*) for novices ($p < 0.05$). Conversely, we find experts given xAI-based support have significantly increased time to build compared to those without xAI-based support, indicating a performance decrease ($t(33) = 2.09; p < 0.05$). We provide further discussion in Section 6.

**Q3:** Expert performance significantly differs across policy information levels ($F(1, 15) = 5.27; p < 0.05$). We find that experts working with cobots augmented with partial information (only the cobot's low-level policy) were able to complete the human-machine teaming task significantly quicker than users with complete information of the cobot's hierarchical policy ($p < 0.05$). We find no significant difference in novice performance across information levels (**IV2**).

**Subjective Findings:** While experts and novices interacted differently with cobots, the subjective findings were similar and are presented as an aggregate. Here, we report on subjective measures that yielded statistical significance. We provide a full analysis within the supplementary material.

**Q2:** We find that users assess positive teammate traits, team working alliance, closeness, and perceived intelligence significantly differently across xAI-based support ($F(2, 56) = [3.16, 4.08, 7.29, 5.31]; [p < 0.05, p < 0.05, p < 0.01, p < 0.01]$). Users rate a cobot with decision-tree xAI-based support as significantly more positive than a cobot with no explanation ($p < 0.05$) and rate human-machine teams with decision-tree xAI-based support with higher working alliance scores than human-machine teams without xAI-based support ($p < 0.05$). We also find that users perceive cobots with decision-tree xAI-based support as significantly more close than cobots without xAI-based support ($p < 0.01$) and cobots with status xAI-based support as significantly more close than cobots with no xAI-based support ($p < 0.05$). In assessing the perceived intelligence of the cobot, we conduct a Friedman's test and find that users perceive cobots with decision-tree xAI-based support as significantly more intelligent than cobots with no explanation ($p < 0.01$). We provide a depiction of our results for **Q2** in Figure 4.

**Q3:** When displaying partial information **IV2: Condition 2**, we find that users trust cobots and rate cobots with higher improvement scores across xAI-based support ($F(2, 28) = [3.55, 5.14]; [p < 0.05, p < 0.05]$). We find users trust cobots augmented with decision-tree xAI-based support in comparison to a cobot augmented with no xAI-based support ($p < 0.05$) and human-machine teams consisting of a cobot with decision-tree xAI-based support are rated with higher improvement scores than human-machine teams consisting of cobots with no xAI-based support ($p < 0.05$).

## 6   Discussion

In Study 1 (Section 4), we start by exploring if xAI techniques can support situational awareness in human-machine teaming. We find objectively that xAI techniques can support SA. Specifically,

we see that providing users with status explanations supports the ability to comprehend AI behavior (Level 2, $p < 0.05$). More so, decision-tree based support provides users with the ability to both comprehend AI behavior (Level 2, $p < 0.001$) and project the cobot's behavior into the near future (Level 3, $p < 0.05$). These results provide promising evidence supporting the deployment of xAI-based support in human-machine teaming.

In Study 2 (Section 5), we assess how cobots augmented with xAI-based support subjectively and objectively affect ad hoc human-machine teaming performance. Novices benefit from a cobot augmented with status xAI-based support ($p < 0.05$) but do not benefit similarly from a cobot augmented with decision-tree xAI-based support. The benefit achieved through status explanations (*SA1-2*) indicates that novices are able to use the support to develop a shared mental model that benefits them. However, the more cognitively intense decision-tree xAI-based support does not provide any benefit, noting that such an xAI technique is not suitable for novices. Expert performance, on the other hand, degrades with the addition of xAI-based support, suggesting that the support serves as a distraction. As experts are more experienced with the game of Minecraft, we hypothesize that the cost of paying attention to the xAI-based support outweighs the benefits obtained from generating an accurate shared mental model. As the cobot is programmed to be collaborative, the cobot behavior may already fall reasonably within the expert's shared mental model, and the support, while providing an accurate description of the cobot's policy, may reduce the user performance. This hypothesis is supported by the further reduction in performance when experts are presented with complete xAI-based support as opposed to partial (**IV2**, $p < 0.05$). Thus, xAI-based support may not be universally beneficial and depends on the composition of the team.

While the performance benefits vary across experts and non-experts, we see that all users find cobots with decision-tree xAI-based support to be more trustworthy, capable of learning, maintain more positive teammate traits, maintain a better working alliance, and be perceived as more intelligent than cobots without xAI-based support. Given the results of our study, we wish to provide some key takeaways for future research in xAI and Human-Machine Teaming (HMT): 1) The addition of xAI techniques can induce SA, an important element of the OODA loop, 2) It is important to account for heterogeneity within both the cobot design [12] and the xAI-based support, and 3) Information overload can exist as an impedance in human-machine teaming and support strategies should consider the cognitive bandwidth of humans.

**Limitations –** In our teaming scenario, the cobot has specific capabilities and does not have equal capability to the human. Our findings are limited to teaming scenarios with similar heterogeneous agents. Our experiment is limited to two broad classes of interpretable models, representing xAI approaches that can generate decision trees and those that can generate a short phrase representing policy information. These abstractions can be generated via a number of methods, including [40, 50]. We also did not explicitly vary the explanation size (e.g., depth of the decision tree, specificity of cobot status, etc.) in our experiment. Increasing the size/complexity of xAI-based support and accordingly modifying the training time is an important future direction to explore, as there is a tradeoff between the utility of xAI-based support and its complexity.

## 7 Conclusion

In this paper, we present findings relating xAI and situational awareness in human-machine teaming. Within our first study, we find that cobots with xAI-based support can provide human teammates a higher level of SA, benefiting a human teammate's ability to perform situational analysis and understand the human-machine teaming scenario. Within our second study, we find that novices working with cobots augmented with status xAI-based support gain significant performance benefits ($p < 0.05$) within the ad hoc human-machine teaming scenario compared to cobots without xAI-based support. Expert performance, on the other hand, degrades with the addition of xAI-based support ($p < 0.05$), indicating that the cost of paying attention to the xAI-based support outweighs the benefits obtained from generating an accurate shared mental model. These results display the benefits and drawbacks of deploying xAI in ad hoc human-machine teaming and provide interesting future directions for xAI-based support in human-machine teaming.

**Broader Impact –** Our work has the potential to benefit future research in deploying xAI in human-machine teaming and advance the envisage of democratizing cobots, focusing on developing higher levels of understanding between humans and cobots. We note that it is important to account for the differences across humans, such as expertise. In other scenarios, it may be important to take into account other demographic information.

## Acknowledgments and Disclosure of Funding

This work was sponsored by MIT Lincoln Laboratory grant 7000437192, NASA Early Career Fellowship grant 80HQTR19NOA01-19ECF-B1, and a gift to the Georgia Tech Foundation from Konica Minolta, Inc.

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
