# The Utility of Explainable AI in Ad Hoc Human-Machine Teaming Supplmentary

**Rohan Paleja**[1], **Muyleng Ghuy**[1], **Nadun R. Arachchige**[1], **Reed Jensen**[2], **Matthew Gombolay**[1]
[1]Georgia Institute of Technology, [2]MIT Lincoln Laboratory
[1]Atlanta, GA 30332, [2]Lexington, MA 02420
{rpaleja3, mghuy3, nkra3}@gatech.edu, rjensen@ll.mit.edu mgombolay3@gatech.edu

## A   Cobot Policy

The cobot maintains a hierarchical policy. A low-level policy determines the cobot's action, and a high-level policy determines the cobot's inference of the human's action (which the low-level policy is conditioned upon). Both components of the hierarchical policy are decision tree-based policies of depth two and with four leaf nodes. The cobot's high-level policy stays fixed throughout gameplay and is displayed in Figure 9. The cobot's low-level policy changes throughout gameplay across five phases of play, correlating with different periods of the task (i.e., building different portions of the house). A depiction of each low-level policy in displayed in Figures 4-8. Across all policies, the cobot has the ability to:

- Chop Birch Wood
- Chop Jungle Wood
- Mine Andesite Stone
- Mine Cobblestone
- Put resources in chest
- Craft pickaxe
- Craft axe
- Craft sticks
- Craft planks

The cobot's behavior loop is to observe the human behavior and perform inference of the human's action. Following the inference of the human's behavior, the cobot determines its action based on the current tree-based low-level policy. Once the action has been completed, the cobot will deposit its excess resources into the chest and begin the loop again.

The cobot deposits any materials that have a quantity above two into the chest for the human to use for building/crafting and any tools with a quantity above one into the chest. The cobot's movement policy (contained within each macro-action) is to move in a straight-line path towards the object. If the agent meets an obstacle and/or takes longer than 9 seconds to reach its goal, the cobot will teleport to the object. For "Chop" or "Mine" actions, the cobot will perform the chopping/mining action until each block of the target has disappeared (detected using Line of Sight variables).

The human teammate is notified in advance that the cobot can teleport if it is having trouble reaching its destination and that the cobot will place extra resources into the chest for the human to retrieve.

### A.1   Status xAI-based Support Display

Here, we provide an image of the status xAI-based support display. In Figure 1, we display the possible status displays of robot actions. In Figure 2, we display the possible status displays of the

cobot inference of the human action. In Figure 3, we provide a visualization of a sample status xAI-based support display.

| | | | |
|---|---|---|---|
| Craft Pickaxe | Mine Andesite Stone | Chop Birch Wood | Craft Sticks |
| Chop Jungle Wood | Craft Axe | Mine Cobblestone | Craft Birch Planks |

Figure 1: Possible Cobot Behavior Displays through Status xAI-based Support

| | | | |
|---|---|---|---|
| Human is Building | Human is Mining | Human is Chopping | Human is Crafting |

Figure 2: Possible Inference of Human Behavior Displays through Status xAI-based Support

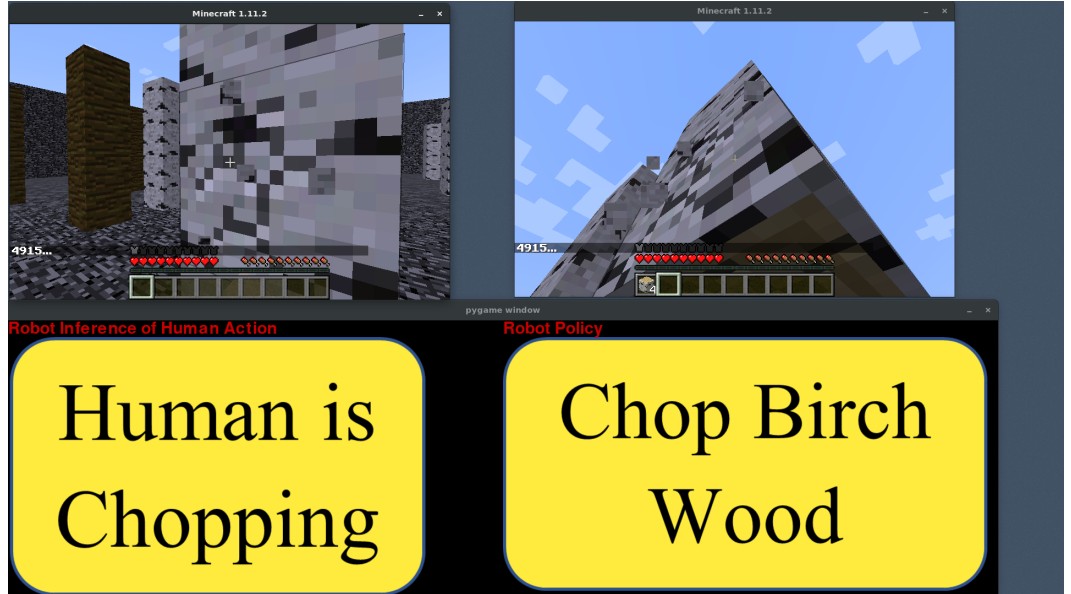

Figure 3: Sample Gameplay Display during Status xAI-based Support

## A.2  Decision-Tree xAI-based Support Display

Here, we provide an image of the decision-tree xAI-based support display alongside all cobot policies. In Figure 4-8, we display the cobot policies throughout gameplay. Each policy corresponds to a different phase of the game determined by a heuristic that assesses current resources and build progress. In Figure 9, we display the decision tree the cobot uses for the inference of the human action. In Figure 10, we provide a visualization of a sample decision xAI-based support display.

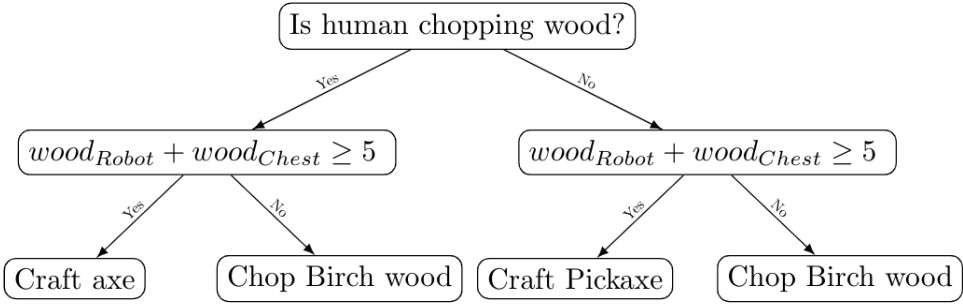

Figure 4: Cobot Behavior Policy During Phase 1

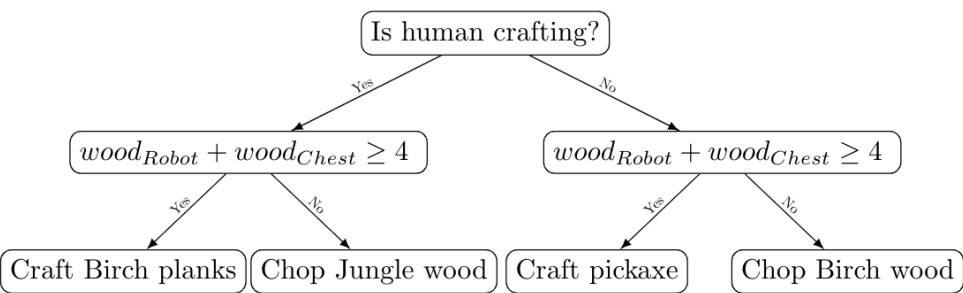

Figure 5: Cobot Behavior Policy During Phase 2

## B Environment Details

Within the environment, there are two types of wood blocks and two types of stone blocks, a crafting table, and a chest. Each resource is scattered throughout the world with clusters in certain regions. The crafting table and chest are at a fixed location. In the area where the human teammate must build the house, colored blocks are placed on the ground to guide the human during the building process. Each color corresponds to a different object that must be built upon the colored block.

The human and cobot are assigned to build a house with certain specifications. The 4-level house contains eight unique objects including 16 Birch planks, 16 Andesite stone, 18 Jungle planks, 18 Cobblestone, 5 stairs, 2 doors, 2 fence gates, and 10 fence pieces, totaling 89 objects. The latter four items are crafted materials that require combining base materials and can be made of any type of wood. A depiction of the 4-level house is displayed in Figure 11.

## C Study 1: Relationship Between Explanations and Situational Awareness Supplementary Material

In this section, we present additional details about the questions maintained in our study and additional analysis details.

### C.1 Study 1: Additional Analysis Procedure

This section provides additional details about the procedure of Study 1 (Section 4).

The experiment was conducted through an online platform where the first page provided the participants with introductory information regarding the human-machine teaming task. Participants are informed that the survey will take approximately one hour, the experiment is completely voluntary, and that the participants will be compensated $20 for the study. Users will then conduct three episodes in which each episode corresponds to a different factor. Each episode consists of ≈10 minutes of

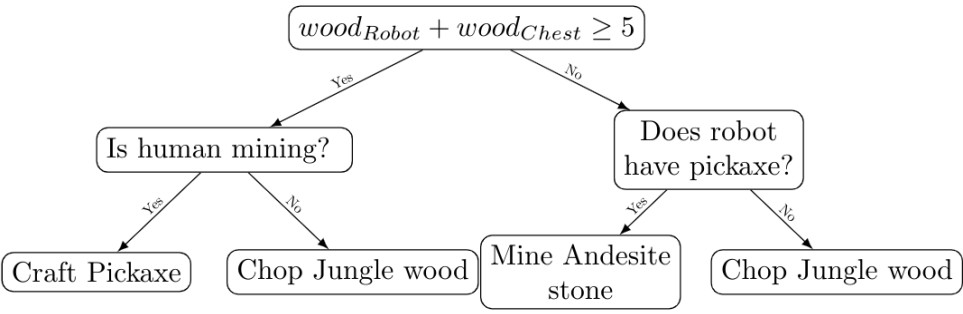

Figure 6: Cobot Behavior Policy During Phase 3

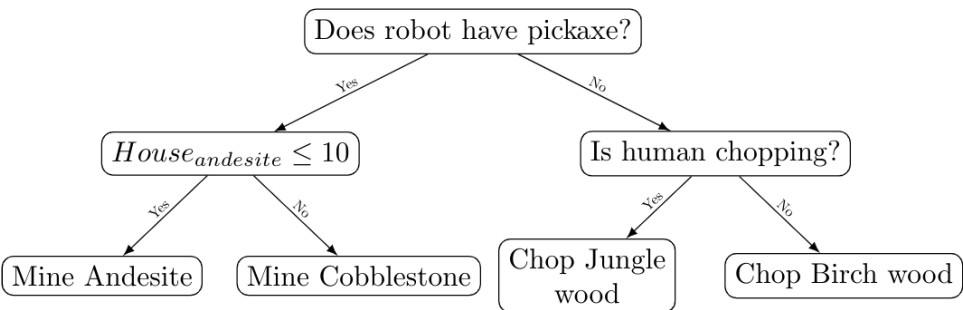

Figure 7: Cobot Behavior Policy During Phase 4

gameplay that users must view interrupted by the SAGAT at two-minute intervals (five trials within each episode). The factor/episode ordering was randomized to mitigate confounds.

### C.2 Study 1: Additional Analysis Details

**Q1:** Overall, we find that xAI techniques support higher levels of situational awareness. We test for normality and homoschedascity across Level 1 and Level 3 SA and do not reject the null hypothesis in either case, using Shapiro-Wilk ($p > 0.1$, $p > 0.2$) and Levene's Test ($p > 0.1$, $p > 0.8$).

We refer the reader to Section 4.3 for the conclusions from the conducted analysis. An additional conclusion not presented within the paper is the difference in Level 2 SA between decision-tree xAI-based support and status xAI-based support. It is found that decision-tree xAI-based support provides users with a higher Level 2 SA compared to those with status xAI-based support ($p < 0.05$)

### C.3 Study 1: Complete List of Situational Awareness Questions

We present the complete list of questions at the end of the supplementary material. We note that the question numbers and page numbers are internal and can be disregarded. We also note that the complete list of answer choices is present in the attached material. However, the questions within the survey are limited to four answer choices by using a randomized answer choices procedure.

## D Study 2: Situational Awareness in Ad Hoc Human-Machine Teaming Supplementary Material

This section provides additional details about Study 2 (Section 5). We provide three sample gameplay clips corresponding to different levels of xAI-based support within the .zip. We note that the gameplay videos are compressed to fit within the size limit of the supplementary material and as a result, there may be slight blurring.

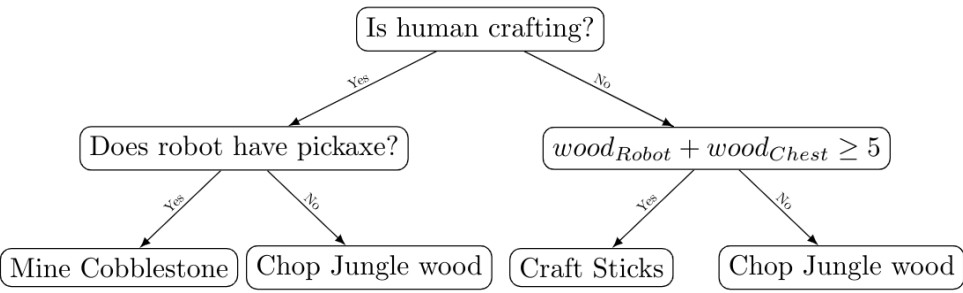

Figure 8: Cobot Behavior Policy During Phase 5

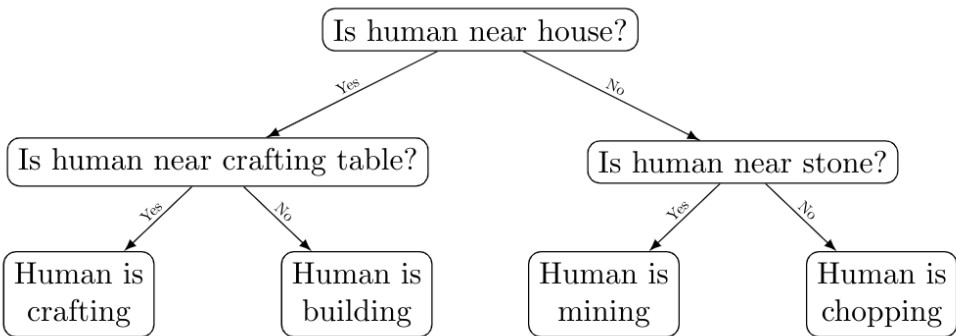

Figure 9: Policy for Human Behavior Inference

## D.1 Study 2: Human-Cobot Teaming Procedure

Participants are first informed that the experiment will take approximately 1.75 hours, the experiment is completely voluntary, and that the participants will be compensated $10 per hour of the study. The participant is also briefed on the objective of the experiment: to investigate the impact of online explanations in human-machine teaming. The participant is told that she will be playing with three *different* autonomous agents (use of deception). The participant starts with a pre-experiment survey collecting demographic information, gaming background (experience with video games and Minecraft), and the Big Five Personality Questionnaire [3]. Afterwards, the participant is handed a instructional document regarding general Minecraft gameplay controls, a crafting guide, specifications for the house and some notes regarding the cobot's behavior. The participant is told that the cobot will place extra resources into the chest for the human and that the cobot will not help the human build. The participant is also informed on how to share tools with the cobot and of all possible cobot behaviors. Then, the participant conducts a hand-crafted tutorial level in Minecraft where participants practice collecting resources, crafting, and placing objects, each a key element within the human-machine teaming task. Per our IRB Protocol, we have a dismissal criterion for the tutorial level to ensure that participants have sufficient knowledge of the game of Minecraft. If participants take longer than ten minutes within the tutorial level, it is deemed they do not have basic knowledge of Minecraft including understanding the controls, placing and crafting objects in the game, etc. We note that within the tutorial, the cobot is present within the environment but is stagnant.

Once completed, the participant is tasked with individually building the entire house. Again, within this domain, the cobot will be present but is stagnant. The individual house build helps the human gain familiarity with the house specifications, which benefits in the user's task understanding (most participants did not need to reference the house specifications handout after the individual house build).

Once completed, the primary experimentation begins. Users will conduct three episodes in which she will play with three cobots, each of which are programmed the same but with varying xAI-

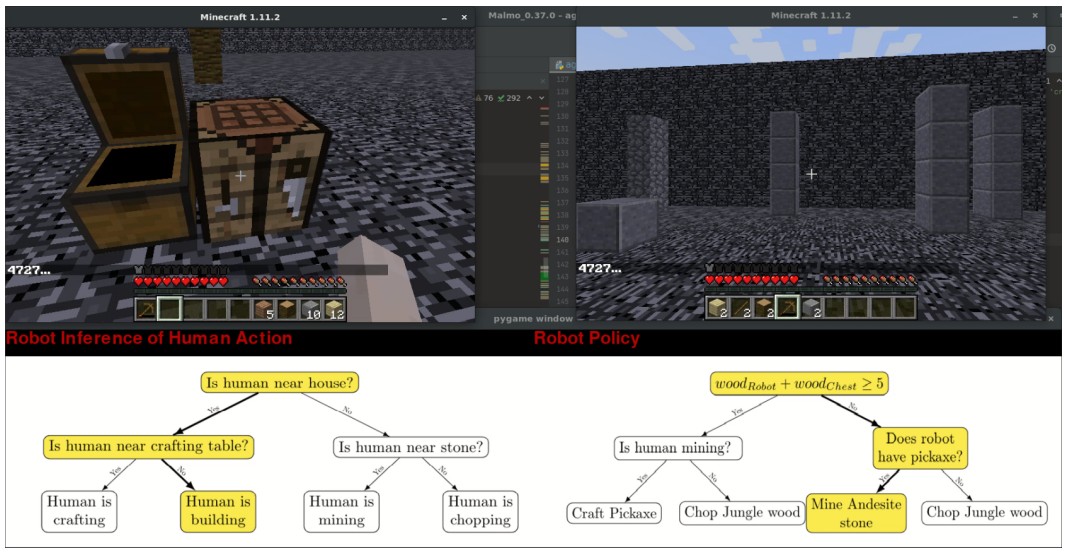

Figure 10: Sample Gameplay Display during Decision-Tree xAI-based Support

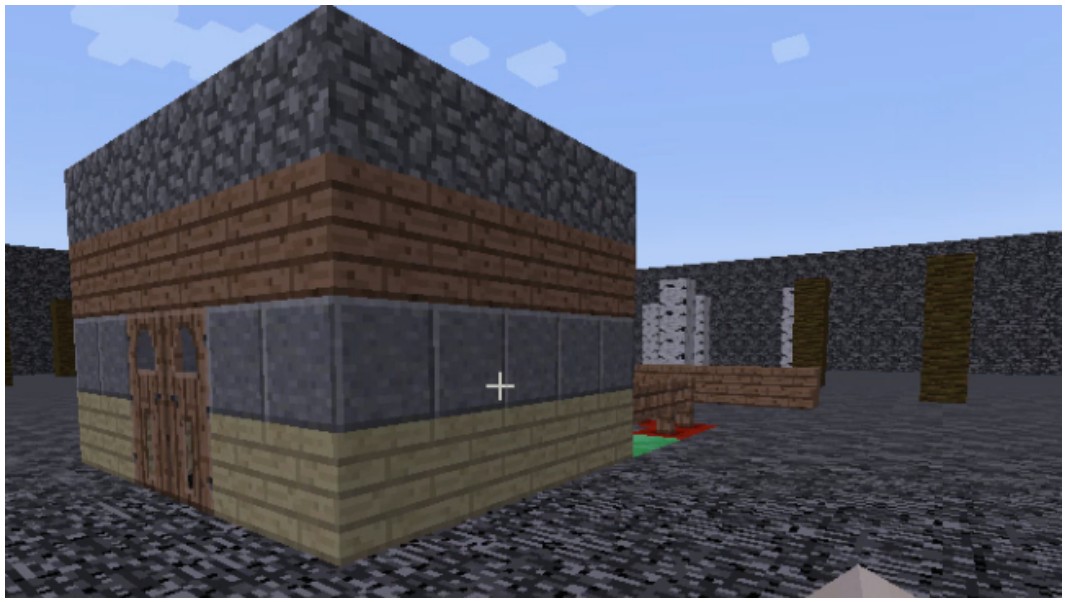

Figure 11: Completed Image of Human-Machine Collaborative Task

based support and a randomly selected level of policy information. The ordering across xAI-based support levels is randomized to mitigate potential experimental confounds, such as learning effect, fatigue, etc. Note that while each cobot is programmed with the same policy, variations in the human's behavior can cause cobot behaviors to widely differ across runs. In each episode, the participant will first be given a sample ≈30-second video that describes the upcoming visualization. Following the video, the participant is given a written tutorial describing the xAI-based support[1]. Once completed, the participant will build the specified house in Minecraft with the cobot, a timed task. To conclude the episode, we administer several post-study scales to support our quantitative findings including [1, 2, 4, 5]. Following the four post-experiment questionnaires, we conduct a semi-structured interview with the participant, asking questions about the user's experience with the cobot. Afterwards, we debrief the participant on the true nature of the study, informing them

---

[1]The *SA1* condition does not have a written tutorial. The *SA1-2-3* written tutorial is augmented with proficiency questions to ensure to user understands how to read decision trees.

that each cobot was programmed the same and that we are studying the effects of different policy abstractions on human-machine teaming performance.

## D.2    Study 2: Additional Analysis Details

Assessing a human-machine team's time-to-build, we test for normality and homoschedascity and do not reject the null hypothesis in either case, using Shapiro-Wilk ($p > 0.05$) and Levene's Test ($p > 0.7$). We find a significant effect between a participant's teaming ability and the participant's build speed ($F(1, 26) = 23.5; p < 0.001$). Thus, we conduct a meta-analysis on our participant data and find two distinctive clusters in participants' individual build times, implying two distinct categories of participants that vary in gameplay speed. We dichotomize our subject data with an additional categorical variable within our analysis representing each cluster, the first referring to "experts" (i.e., those with higher proficiency in the game of Minecraft) and "novices" (i.e., those with lower proficiency in the game of Minecraft). Within our model, we utilize several covariates including demographic information (gender, age), gaming familiarity, and personality in determining which factors are significant. Within each category of the expertise variable, for novices and experts, we test for normality and homoscedasticity and do not reject the null hypothesis in either case, using Shapiro-Wilk ($p > 0.2$, $p > 0.1$) and Levene's Test ($p > 0.8$, $p > 0.9$).

We refer the reader to Section 5.3 for the conclusions from the conducted analysis.

### D.2.1    Study 2: Subjective Analysis Details

Here, we look through each post-study scale and provide additional details regarding analysis and conclusions.

**Human-Robot Collaborative Fluency Assessment** Here, we present the findings and Cronbach alpha scores for each sub-scale within the human-robot collaborative fluency assessment. For each measure, we maintain a 7-point response format per item within the Likert scale.

- Human-Robot Fluency – We do not find any significance across different xAI-based support levels IV1 and policy information levels IV2. We note that we maintain a Cronbach's $\alpha$ of 0.79 for this scale.

- Robot Relative Contribution – We do not find any significance across different xAI-based support levels IV1 and policy information levels IV2. We note that we maintain a Cronbach's $\alpha$ of 0.76 for this scale.

- Trust in Robot – We test for normality and homoschedascity and do not reject the null hypothesis in either case, using Shapiro-Wilk ($p > .70$) and Levene's Test ($p > 0.05$). We find a significant difference in teammate trust across xAI-based support levels ($F(2, 28) = 3.55; p < 0.05$). We find that when only the explanation of the robot policy is available, the AI with a decision-tree xAI-based support is perceived as more trustworthy than an AI without xAI-based support ($p < .05$). We note that we maintain a Cronbach's $\alpha$ of 0.81 for this scale.

- Positive Teammate Traits – We test for normality and homoschedascity and do not reject the null hypothesis in either case, using Shapiro-Wilk ($p > .15$) and Levene's Test ($p > 0.05$). We find a significant difference in positive teammate traits across xAI-based support levels ($F(2, 56) = 3.16; p < 0.05$). In the post-hoc analysis, we find that a cobot with decision-tree xAI-based support is rated with significantly higher positive teammate traits than an cobot without explanation ($p < .05$). We note that we maintain a Cronbach's $\alpha$ of 0.78 for this scale.

- Improvement – We test for normality and homoschedascity and do not reject the null hypothesis in either case, using Shapiro-Wilk ($p > .80$) and Levene's Test ($p > 0.05$). We find a significant difference in perceived improvement across xAI-based support levels ($F(2, 28) = 5.14; p < 0.05$). In the post-hoc analysis, we find that a cobot with decision-tree xAI-based support is rated with significantly higher positive teammate traits than an cobot without explanation ($p < .05$). We note that we maintain a Cronbach's $\alpha$ of 0.85 for this scale.

- Working Alliance for Human-Robot Teams – We test for normality and homoschedascity and do not reject the null hypothesis in either case, using Shapiro-Wilk ($p > .10$) and

Levene's Test ($p > 0.80$). We find a significant difference between the team working alliance across different xAI-based support levels ($F(2, 56) = 4.08; p < 0.05$). We find that decision-tree xAI-based support is rated with a higher working alliance than no xAI-based support ($p < 0.05$). We note that we maintain a Cronbach's $\alpha$ of 0.89 for this scale.

- Inclusion of the Other in the Self – We test for normality and homoschedascity and do not reject the null hypothesis in either case, using Shapiro-Wilk ($p > .55$) and Levene's Test ($p > 0.75$). We find a significant difference in perceived closeness across xAI-based support levels ($F(2, 56) = 7.29$); $p < 0.01$). In the post-hoc analysis, we find that users perceive cobots with decision-tree xAI-based support as significantly more close than cobots without xAI-based support ($p < 0.01$) and cobots with status xAI-based support as significantly more close than cobots without xAI-based support ($p < 0.05$).

**Godspeed Questionnaire** Within this subsection, we look at two different measures within the Godspeed Questionnaire [2], likability and perceived intelligence. For each measure, we maintain a 5-item Likert scale with a 5-point response format.

- Likability – We do not find any significance across different xAI-based support levels IV1 and policy information levels IV2. We note that we maintain a Cronbach's $\alpha$ of 0.94 for this scale.
- Perceived Intelligence – A Friedman's test found significance ($\chi^2(2) = 8.02; p < 0.05$) for perceived intelligence across different levels of xAI-based support. The pairwise analysis finds that decision-tree xAI-based support ($p < 0.05$) significantly increases perceived intelligence in comparison to a cobot without xAI-based support. We note that we maintain a Cronbach's $\alpha$ of 0.88 for this scale.

**NASA-TLX Workload Survey** We do not find any significance across different xAI-based support levels IV1 and policy information levels IV2. We note that we maintain a Cronbach's $\alpha$ of 0.68 for this scale.

# E   Study 1: Complete List of Situational Awareness Questions

Here, we provide a complete list of questions for measuring Level 1, Level 2, and Level 3 SA used in Study 1: Relationship between Explanations and Situational Awareness. To minimize the interruption of the SAGAT, questions are multiple-choice and are sampled from the set of questions below (assessing the user's complete SA would require a very large number of questions).

QID25 **Level 1**

- - - - - - - - - - - - - - - - - - - - - - - - - - - - - - - - - - - - - - - - - - - - - - -

- - - - - - - - - - - - - - - - - - - - - - - - - - - - - - - - - - - - - - - - - - - - - - -

QID21 What is the **human** currently doing?

   ◯ Crafting  (13)

   ◯ Building  (14)

   ◯ Mining  (15)

   ◯ Chopping  (16)

- - - - - - - - - - - - - - - - - - - - - - - - - - - - - - - - - - - - - - - - - - - - - - -

QID48 What is the **robot** currently doing?

   ◯ Chopping Birch Wood  (1)

   ◯ Mining Andesite Stone  (2)

   ◯ Mining Cobblestone  (3)

   ◯ Crafting axe  (4)

   ◯ Crafting pickaxe  (5)

   ◯ Crafting  planks  (6)

   ◯ Chopping Jungle Wood  (7)

   ◯ Crafting sticks  (8)

   ◯ None of the above  (9)

- - - - - - - - - - - - - - - - - - - - - - - - - - - - - - - - - - - - - - - - - - - - - - -

QID27 How many pieces of wood are currently in the **chest**? Please add the number of birch and jungle wood. Note this does not include planks.

○ 0-2  (1)

○ 3-5  (2)

○ 6-8  (3)

○ 9 or greater  (4)

---

QID45 How many pieces of wood does the **human** currently have? Please add the number of birch and jungle wood. Note this does not include planks.

○ 0-2  (1)

○ 3-5  (2)

○ 6-8  (3)

○ 9 or greater  (4)

---

QID46 How many pieces of wood does the **robot** currently have? Please add the number of birch and jungle wood. Note this does not include planks.

○ 0-2  (1)

○ 3-5  (2)

○ 6-8  (3)

○ 9 or greater  (4)

---

QID50 How many pieces of stone are currently in the chest? Please add the number of andesite stone and cobblestone.

○ 0-2  (1)

○ 3-5  (2)

○ 6-8  (3)

○ 9 or greater  (4)

---

QID51 How many pieces of stone does the human currently have? Please add the number of andesite stone and cobblestone.

○ 0-2  (1)

○ 3-5  (2)

○ 6-8  (3)

○ 9 or greater  (4)

---

QID52 How many pieces of stone does the **robot** currently have? Please add the number of andesite stone and cobblestone.

○ 0-2  (1)

○ 3-5  (2)

○ 6-8  (3)

○ 9 or greater  (4)

---

QID53 How many birch wood planks have been placed onto the house?

○ 0-3  (1)

○ 4-7  (2)

○ 8-11  (3)

○ 12 or greater  (4)

---

QID55 How many andesite stone blocks have been placed onto the house?

○ 0-3  (1)

○ 4-7  (2)

○ 8-11  (3)

○ 12 or greater  (4)

---

QID56 How many jungle wood planks have been placed onto the house?

○ 0-3  (1)

○ 4-7  (2)

○ 8-11  (3)

○ 12 or greater  (4)

---

QID57 How many cobblestone blocks have been placed onto the house?

○ 0-3  (1)

○ 4-7  (2)

○ 8-11  (3)

○ 12 or greater  (4)

QID26 **Level 2**

---

---

QID58 It is possible for the robot to chop birch wood with its current decision-making strategy.

○ True  (1)

○ False  (2)

QID69 It is possible for the robot to chop jungle wood with its current decision-making strategy.

○ True  (1)

○ False  (2)

QID70 It is possible for the robot to mine andesite stone with its current decision-making strategy.

○ True  (1)

○ False  (2)

---

QID72 It is possible for the robot to craft an axe with its current decision-making strategy.

○ True  (1)

○ False  (2)

---

QID73 It is possible for the robot to craft a pickaxe with its current decision-making strategy.

○ True  (1)

○ False  (2)

---

QID74 It is possible for the robot to craft sticks with its current decision-making strategy.

○ True  (1)

○ False  (2)

---

QID75 It is possible for the robot to craft planks with its current decision-making strategy.

○ True  (1)

○ False  (2)

---

QID71 It is possible for the robot to mine cobblestone with its current decision-making strategy.

○ True  (1)

○ False  (2)

---

QID68 The robot utilizes information about the amount of wood the robot is holding while making a decision.

○ True  (1)

○ False  (2)

---

QID64 The robot utilizes information about the amount of andesite stone on the house while making a decision.

○ True  (1)

○ False  (2)

---

QID62 The robot utilizes information about the amount of wood in the chest while making a decision.

○ True  (1)

○ False  (2)

---

QID59 The robot utilizes information about whether the human is crafting while making a decision.

○ True  (1)

○ False  (2)

QID63 The robot utilizes information about whether it has a pickaxe while making a decision.

○ True  (1)

○ False  (2)

QID60 The robot utilizes information about whether the human is chopping while making a decision.

○ True  (1)

○ False  (2)

QID61 The robot utilizes information about whether the human is mining while making a decision.

○ True  (1)

○ False  (2)

QID66 The robot utilizes information about what the human is doing while making a decision.

○ True  (1)

○ False  (2)

QID65 The robot utilizes information about whether the human is building while making a decision.

○ True  (1)

○ False  (2)

QID29 **Level 3**

- - - - - - - - - - - - - - - - - - - - - - - - - - - - - - - - - - - - - - - - - - - - - - - - - - - - -

- - - - - - - - - - - - - - - - - - - - - - - - - - - - - - - - - - - - - - - - - - - - - - - - - - - - -

⤨

QID31 Assume the chest has 10 pieces of wood. As the human player, what would you do to coerce the robot to chop any type of wood?

○ Go mine stone  (1)

○ Go chop wood  (2)

○ Go to build the house  (3)

○ Go to craft items  (4)

○ The robot's policy does not depend on my behavior.  (5)

○ It does not matter what I do. The robot will not pursue this behavior.  (6)

○ None of the above.  (7)

- - - - - - - - - - - - - - - - - - - - - - - - - - - - - - - - - - - - - - - - - - - - - - - - - - - - -

⤨

QID77 After the robot completes its current action, assuming the human repeats its current action, what will the robot do next?

○ Chop Birch Wood  (1)

○ Mine Andesite Stone  (2)

○ Mine Cobblestone  (3)

○ Craft axe  (4)

○ Craft pickaxe  (5)

○ Craft  planks  (6)

○ Chop Jungle Wood  (7)

○ Craft sticks  (8)

○ None of the above.  (9)

---

QID32 You would like the robot to mine cobblestone. What input requirements must hold true for the robot to perform such a behavior?

○ More than 10 wood in the chest, human needs to be mining stone.  (3)

○ Less than 4 wood among the chest and robot's inventory, the human needs to be crafting.  (4)

○ The andesite layer of the house has not been started, and the robot needs to have a pickaxe.  (5)

○ The robot needs a pickaxe, and the human must be crafting.  (6)

○ None of the above.  (7)

---

QID78 You would like the robot to chop birch wood. What input requirements must hold true for the robot to perform such a behavior?

○ More than 10 wood in the chest, human needs to be mining stone. (3)

○ Less than 4 wood among the chest and robot's inventory, the human needs to be crafting. (4)

○ More than 10 wood in the chest, human needs to be chopping wood. (5)

○ The robot needs a pickaxe, and the human must be crafting. (6)

○ None of the above. (7)

---

QID80 You would like the robot to chop jungle wood. What input requirements must hold true for the robot to perform such a behavior?

○ More than 10 wood in the chest, human needs to be mining stone. (3)

○ Less than 4 wood among the chest and robot's inventory, the human needs to be crafting. (4)

○ Less than 4 wood among the chest and robot's inventory, the robot does not have a pickaxe (5)

○ Human is chopping wood. The robot does not have a pickaxe. (6)

○ None of the above. (7)

---

QID79 You would like the robot to craft a pickaxe. What input requirements must hold true for the robot to perform such a behavior?

○ More than 10 wood in the chest, human needs to be mining stone.  (3)

○ Less than 4 wood among the chest and robot's inventory, the human needs to be crafting.  (4)

○ More than 10 wood in the chest, human needs to be chopping wood.  (5)

○ The robot needs a pickaxe, and the human must be crafting.  (6)

○ None of the above.  (7)

End of Block: Level 3