# OpenReview forum: "The Utility of Explainable AI in Ad Hoc Human-Machine Teaming"
_NeurIPS.cc/2021/Conference — NeurIPS 2021 Poster_

### Official Review · Reviewer_VynQ · 2021-07-08

**Rating:** 5
**Confidence:** 2

**Summary:**

This paper makes an initial study on how explanations from an AI agent impact human productivity on Minecraft tasks.


**Ethical Concerns:**

The only concern I thought of was the use of human subjects, but it was IRB approved.


**Limitations And Societal Impact:**

Yes

**Main Review:**

The problem studied in this paper is an important one, given the extensive amount of explainability work in the field right now.  The paper is also well-written and uses sound statistical analysis methods.

I am concerned about whether the narrow scope of the experiments makes this paper a poor fit to this conference.  Their results are understandable that different explanation models will have different effects, particularly if information overload might slow an expert on certain tasks.  However, it seems reasonable that other explanation approaches might work much better, improving productivity.  E.g., what of an image-based explanation (with an easily read color map), which could be read at a glance, for an expert?  Or what of a natural language-based explanation (more tutorial-like) for a novice?  Or a model that adapts its explanations based on human performance?  It seems that this question is better studied in tandem with HCI.

Another question I have is how much does the quality of the agent's policy matter?   E.g., what if the cobot is dumb, or at least less optimal?   What effect do the different types of explanations have then?

Detailed comments;
- Line 78: "a ad hoc" -> "an ad hoc"
- Line 164: "find-tune"
- In Section 4.1, it would be good to have a citation for SAGAT.
- In Section 5.2, why the use of deception?
- Line 387: "findings is limited"


**Time Spent Reviewing:**

2

---

> ### Author Response · Authors · 2021-08-10
> **Reviewer VynQ Response**
>
> We thank you for your positive feedback regarding our problem, clarity, and sound analysis.
>
> **Narrow Scope of the Experiments**
>
> Text-based and decision-tree based policy explanations are two common types of explanations. A large number of frameworks can generate decision trees and text-based descriptions from a policy. Other explanations such as image-based (colormap), NLP-generated, and adaptive may prove useful in future work. We hope that our paper provides initial guiding principles in the design of future xAI techniques and motivates further studies with different types of explanations.
>
> **Suboptimal Agent Policy**
>
> Policy explanations when the cobot is suboptimal (where the policy explanation is accurate while policy itself is suboptimal) is beyond the scope of this paper. We pose this as an interesting future direction. Prior work (Zhang et al) has presented some insight into suboptimal AI models for recommender systems in healthcare, finding that suboptimality reduces the trust in the AI.
>
> Zhang, Zhan et al. “Effect of AI Explanations on Human Perceptions of Patient-Facing AI-Powered Healthcare Systems.” Journal of Medical Systems 45 (2021): n. pag.

---

> > ### Author Response · Authors · 2021-08-30
> > **Reviewer VynQ Follow Up**
> >
> > Please let us know if we can provide any more information in support of the paper's acceptance. If our rebuttal has addressed your concerns, we kindly ask the reviewer might consider please increasing the reviewer's score and/or confidence? Otherwise, we would be happy to continue discussing any remaining items! Thank you again for your review.

---

### Official Review · Reviewer_jmSk · 2021-07-09

**Rating:** 5
**Confidence:** 4

**Summary:**

The paper presents two user studies to quantify the impact of explanations on Human-AI teaming. The first study experiments with varying cobot (AI) policies to understand how well humans can create a mental model and situational awareness. Results from this study show that explanations can assist mental model creation. The second study instead also explores how human expertise and ability impacts this process. Results from this study show that for novice users there is a clear benefit from XAI but for expert users such benefits degrade due to the extra cost that humans have to pay to process the AI/XAI information.

The formalization of situational awareness in 3 levels (perception, comprehension, projection) is interesting and potentially reusable for other RL applications.

Results are clearly presented and sufficient information is included for clarity and reproducibility.

**Ethical Concerns:**

Authors have adequately touched upon the importance of considering a diverse set of human expertise in such studies and also the potential impact of automation bias and over reliance.

**Limitations And Societal Impact:**

Authors have adequately addressed the limitations and potential negative societal impact of their work.

Further suggestions beyond clarifying the work for novelty, would be to describe how the same type of approach or study style could apply to other RL applications.

**Main Review:**

The questions posed by the paper are important and relevant to the XAI field.

The paper and its novelty however is not very well position with respected work. The question of whether explanations help Human-AI teams has been recently explored in several works which are not discussed in this submission (see list below for examples). A lot of this related work also suggests that benefits from AI and XAI are higher when the human performance without AI is lower than AI alone. A lot of other work has also shown that it is difficult to improve human-ai team performance for experts. Thus, results in the paper need to be contextualized and clarified for novelty. Perhaps some aspect of novelty to consider is the RL application in this case. However, there is still some work in this area as well.




Related Work to consider:
On human predictions with explanations and predictions of machine learning models: A case study on deception detection

Bansal, G., Wu, T., Zhou, J., Fok, R., Nushi, B., Kamar, E., ... & Weld, D. (2021, May). Does the whole exceed its parts? the effect of ai explanations on complementary team performance. In Proceedings of the 2021 CHI Conference on Human Factors in Computing Systems (pp. 1-16).

Gonzalez, A. V., Bansal, G., Fan, A., Jia, R., Mehdad, Y., & Iyer, S. (2020). Human Evaluation of Spoken vs. Visual Explanations for Open-Domain QA. arXiv preprint arXiv:2012.15075.

Adrian Bussone, Simone Stumpf, and Dympna O'Sullivan. 2015. The Role of Explanations on Trust and Reliance in Clinical Decision Support Systems. In Proceedings of the 2015 International Conference on Healthcare Informatics (ICHI '15). IEEE Computer Society, USA, 160–169. DOI:https://doi.org/10.1109/ICHI.2015.26

Effect of AI Explanations on Human Perceptions of Patient-Facing AI-Powered Healthcare Systems.

Zhang, Yunfeng, Q. Vera Liao, and Rachel K. E. Bellamy. “Effect of Confidence and Explanation on Accuracy and Trust Calibration in AI-Assisted Decision Making.” Proceedings of the 2020 Conference on Fairness, Accountability, and Transparency (2020): n. pag. Crossref. Web.

Mental Models of Mere Mortals with Explanations
of Reinforcement Learning

**Time Spent Reviewing:**

2.5

---

> ### Author Response · Authors · 2021-08-10
> **Reviewer jmSK Response**
>
> We thank you for your feedback regarding our clarity and reproducibility.
>
> **Related Work**
>
> We thank the reviewer for pointing us to related work that look at xAI in human-robot teaming. We provide a general description of our novelty with respect to the references you have suggested.
>
> Here, we describe the related work and find that our work is novel with respect to each, providing a valuable contribution to xAI in human-robot teaming literature. Lai et al., Bansal et al., Gonzalez et al., Bussone et al., Zhang et al., and Zhang et al. provide interesting findings relating explanation type and accuracy to trust and reliance in human-AI teaming with explanations. However, while Lai et al., Bansal et al., Bussone et al., Zhang et al., Gonzalez et al., and Zhang et al. deploy xAI-based support in classification problems (utilizing the AI as a transparent recommender system), the task is widely different from our scenario, a complex sequential decision-making problem where the AI is a collaborative agent that actively shapes the world. Anderson et al. provides interesting insight into how different explanations relate to mental model generation in a real-time strategy game. We expand on this by first relating xAI techniques to situational awareness, grounding it in a concept highly important in human-machine teaming. Furthermore, our experiment also provides human teammates with the complete cobot policy in the form of an interpretable decision tree, a type of explanation that is not among the referenced works. Finally, our experiment differs from Anderson et al. in that we assess performance among the human-cobot team with online explanations (Study 2). In this way, users must build mental models online while collaborating with the cobot.
>
> **Further suggestions beyond clarifying the work for novelty, would be to describe how the same type of approach or study style could apply to other RL applications.**
>
> We will add details to how our study style can be quickly generalized to other explanation types and/or other RL applications. We plan to release a detailed codebase so that others can reproduce our experiment and make changes to the policy and explanation type in future work.

---

> > ### Author Response · Authors · 2021-08-30
> > **Reviewer jmSK Follow Up**
> >
> > Please let us know if we can provide any more information in support of the paper's acceptance. If our rebuttal has addressed your concerns, we kindly ask the reviewer might consider please increasing the reviewer's score? Otherwise, we would be happy to continue discussing any remaining items! Thank you again for your review.

---

### Official Review · Reviewer_j9dy · 2021-07-09

**Rating:** 5
**Confidence:** 4

**Summary:**

This work investigates the impact of different levels of robot explanations in the context of human-machine collaboration, with respect to the impact on 1) the human’s situational awareness and 2) the time it takes to complete the task. It studies this in two separate human user studies.

**Limitations And Societal Impact:**

This work addresses some limitations, but does not adequately acknowledge the limitation of whether these results are applicable for more complex policies.

This work has no direct negative societal impacts.

**Main Review:**

**Overall:**

This paper is well-written and clear. The experiments are well-designed and the metrics capture the outcomes of interest. The experiment task itself, collaboratively building a house in Minecraft, is an interesting and non-trivial task. I also appreciate that the human-robot collaboration is unstructured, in that the human is free to choose the extent to which they collaborate with the robot; this makes the studies more ecologically valid.

The results of the study on time completion are surprising and useful: experts perform the task most efficiently with no robot explanations at all, and novices perform best with simple robot explanations (i.e., text) rather than more detailed explanations (i.e., decision trees). This surprising result is of potential interest to the human-robot interaction community.

The main limitation of the work is that the policy is hand-crafted and very simple: a high-level and low-level policy, each of which is a two-level binary decision tree with a total of four leaf nodes. It is unclear whether the results of this study could be generalized for more complex policies and explanations, that cannot be summarized so concisely.

I am also not sure whether this work is a good fit for NeurIPS, given that there is no algorithmic contribution. I think this would be a strong submission for HRI, since the main contribution is the user studies.

**Novelty:**

The main novelty of this work is its thorough investigation of the utility of explainable AI for human-machine collaboration, in the form of two user studies on a non-trivial collaboration task in Minecraft. There is no algorithmic or theoretical novelty.

**Quality:**

The experiments are well-designed. The analysis of the experiments is thorough and statistically sound.

**Clarity:**

This paper is well-written, and overall the motivation and user study design are explained clearly. I just have one clarification question about the first user study—were the status and decision-tree explanations for both the high and low level, or just for the low level?

**Significance:**

I found it interesting that users had the most positive feelings toward the robot with decision-tree explanations, based on their answers to questions about trustworthiness, intelligence, positive teammate traits, etc. But they actually performed worse with it, compared to robots with simpler explanations (that they didn’t feel as positively about). This is a valuable insight for the human-robot interaction community, although it is unclear whether this would generalize to more complex policies and tasks.

In terms of future work, something I am quite curious about is whether there is an explanation technique that would obtain similar positive user perception as the decision-tree explanations, without hurting performance on the task.

**Minor suggestions:**
- In Figures 3b and 3c, it would be good to state what the performance residuals are with respect to. I assume it’s with respect to the no-explanation condition.

**Time Spent Reviewing:**

3

---

> ### Author Response · Authors · 2021-08-10
> **Reviewer j9dy Response**
>
>
> We thank you for your feedback regarding our studies, experiment domain, and structure of collaboration.
>
> **Impact of Increased Size/Complexity Explanations**
>
> We did not explicitly vary the explanation size (depth of the decision tree/specificity of cobot status) in our experiment and will note this as a limitation. Through pilot studies, we augmented the cobot’s behavior and the resulting decision tree size so as to balance the needs to minimize the amount of time participants would need to spend to understand and utilize the tree vs. maximize the ability of the cobot to adapt to and collaborate with the participants. Increasing the size/complexity of xAI-based support and accordingly modifying the training time is an important future direction to explore, as there is a tradeoff between the utility of xAI-based support.
>
> **Not a good fit for NeurIPS**
>
> While our paper does not provide an algorithmic contribution, we believe our paper is a good fit for NeurIPS as it brings to light several interesting findings that can inform the design of future xAI techniques that may be used for support in human-machine teaming. Quoting Professor Michael Carbin from the Charles Isbell’s NeurIPS 2020 keynote, “The issue is not just correctness, but understanding the problem across the entire pipeline to understand what correctness is.”
>
> **Were the status and decision-tree explanations for both the high and low level, or just for the low level?**
>
> Study 1 utilized both the high-level and low-level in its policy explanations.
>
> **In Figures 3b and 3c, it would be good to state what the performance residuals are with respect to. I assume it’s with respect to the no-explanation condition.**
>
> We will clarify this within our paper. The performance residuals are with respect to the no-explanation condition.

---

> > ### Author Response · Authors · 2021-08-30
> > **Reviewer j9dy Follow Up**
> >
> > Please let us know if we can provide any more information in support of the paper's acceptance. If our rebuttal has addressed your concerns, we kindly ask the reviewer might consider please increasing the reviewer's score? Otherwise, we would be happy to continue discussing any remaining items! Thank you again for your review.

---

### Official Review · Reviewer_d8ML · 2021-07-16

**Rating:** 8
**Confidence:** 2

**Summary:**

The authors present two user studies assessing the effect of decision tree-based explanations on human teammates in a human-cobot collaborative task - and specifically a minecraft house building task.  The results show that, while explanations can help situational awareness of the human subject, they can also overwhelm novices and hinder experts.

**Ethical Concerns:**

None.

**Limitations And Societal Impact:**

The authors explicitly address some limitations of their study in Section 6.  Perhaps one could add that the effect of explanation size was not explicitly tested, and that explanations were assumed to be faithful to the policy, which is often not the case in current research.

The broader impact paragraph is just fine.

**Main Review:**

The paper is a pleasure to read.  Measuring the effects of explanations in human-machine collaborative settings is extremely important for all interactive AI/ML applications.  The idea of doing so using a modified Minecraft platform is quite clever.  I could not find any major issues with the experimental design.  The evaluation is quite thorough.

To the best of my knowledge, the related work has good coverage with one exception that I could find:  the potentially harmful effects of machine explanations have been thoroughly explored in

  Ai et al., "Beneficial and harmful explanatory machine learning". MLJ 2021.
  https://link.springer.com/article/10.1007/s10994-020-05941-0

within the context of logic-based explanations.  Their results seem to anticipate the findings about the negative effects of explanations on novice players (although not the ones on expert players, as far as I could see).  It would be fair to mention and briefly discuss the link to this very relevant prior work.

This is a solid and welcome contribution and I do not have any majore complaints about the paper, just one remark/question.

1. The explanations used in the experiments are quite small, essentially conjunctions of two literals.  Did you attempt to asses the impact of increasing the size/complexity of the explanations?  This question is quite central in the current XAI context, in which researchers do not shy away from presenting users with very detailed explanations.

**Time Spent Reviewing:**

4

---

> ### Author Response · Authors · 2021-08-10
> **Reviewer d8ML Response**
>
> We thank the reviewer for the positive feedback regarding our problem, experiment domain, and experiment design. We will include the relevant related work of Ai et al. within our Related Work section.
>
> **Impact of Increased Size/Complexity Explanations**
>
> We did not explicitly vary the explanation size (depth of the decision tree/specificity of cobot status) in our experiment and will note this as a limitation. Through pilot studies, we augmented the cobot’s behavior and the resulting decision tree size so as to balance the needs to minimize the amount of time participants would need to spend to understand and utilize the tree vs. maximize the ability of the cobot to adapt to and collaborate with the participants. Increasing the size/complexity of xAI-based support and accordingly modifying the training time is an important future direction to explore, as there is a tradeoff between the utility of xAI-based support.
>
> **Correctness of Explanation**
>
> We assume that explanations are faithful to the policy and will note this as a potential limitation. Including a binary representation of explanation accuracy as a tertiary variable would result in a 3 by 2 by 2 study, a difficult study to conduct and analyze.  In future, we believe a study solely into the correctness of xAI-based support serves as a very interesting direction.

---

> > ### Author Response · Authors · 2021-08-30
> > **Reviewer d8ML Follow Up**
> >
> > Please let us know if we can provide any more information in support of the paper's acceptance. If our rebuttal has addressed your concerns, we kindly ask the reviewer might consider please increasing the reviewer's confidence? Otherwise, we would be happy to continue discussing any remaining items! Thank you again for your review.

---

### Official Review · Reviewer_jxhE · 2021-07-28

**Rating:** 7
**Confidence:** 5

**Summary:**

This paper presents a study on the effect of two types of interpretable/explainable model in human-machine teaming. By varying the type of explanation (showing the model for human intention recognition used by the robot and showing the robot's action-selection model), the authors conclude that showing both models increased participants' situation awareness, but that only the human intention recognition model decreased the overall task completion time. For 'expert' participants, the performance degraded with


**Limitations And Societal Impact:**

Not particularly. In the 'broader impact' section, the authors mostly just summarise the general benefits of explainability. Looking into some of the broader issues with explainability, particularly in a paper that questions the usefulness for experts in particularly scenarios, would be useful.

**Main Review:**

Related work
============

Despite the claim in the paper, there is a bit of work looking at XAI in human-robot teaming; although it is not directly called this. Some articles I know of:

- Some of Tatagatha Chakraborti's work looks at explanation and explicability for cobots -- there are a number of papers, and some of the tasks are collaborative.

- Neerincx et al. Using perceptual and cognitive explanations for enhanced human-agent team performance. In International Conference on Engineering Psychology and Cognitive Ergonomics (pp. 204-214). Springer, Cham., 2018 -- this paper looks to build collaborative teaming

- Madumal et al. Explainable reinforcement learning through a causal lens. In Proceedings of the AAAI Conference on Artificial Intelligence (Vol. 34, No. 03, pp. 2493-2500), 2020 -- this paper does evaluations with collaborative tasks.

Further, the `status explanation' appears to be the same as the idea of communicating intent. If the leaf nodes of the decision trees are explained, then this is simply explaining what the robot will do next. This has been studied in prior work in human-agent teaming; e.g., there are the articles that I know of, presumably there are others:

- Harbers et al.: Explanation and coordination in humanagent teams: a study in the BW4T testbed. In Proceedings of the 2011 IEEE International Conferences on Web Intelligence and Intelligent Agent Technology, pages 17–20. IEEE Computer Society, 2011.

- Harbers et al. Explanation in human-agent teamwork. In Coordination, Organizations, Institutions, and Norms in Agent System VII, pages 21–37. Springer, 2012.

- Li et al. Communication in human-agent teams for tasks with joint action. In International Workshop on Coordination, Organizations, Institutions, and Norms in Agent Systems (pp. 224-241). Springer, Cham, 2015.

- Unhelkar et al. Decision-making for bidirectional communication in sequential human-robot collaborative tasks. In Proceedings of the 2020 ACM/IEEE International Conference on Human-Robot Interaction (pp. 329-341), 2020.


None of these touch on the same issues outlined in the paper, but they are perhaps more closely related than other references in the related work section.

Studies
=======

The studies using Malmo and the tasks are excellent. The task itself is reasonably rich and do run an interactive colalborative task is difficult to setup. This is the type of evaluation I'd love to see more of, and would love to do more of. I hope the authors share the setup and code for this once this paper is published. The actual policies are reasonably simple; perhaps simpler than anything I've seen in a truly collaborative task, but I do understand the limitations of working with participants recruited for studies and the time limits for training etc.

The study is well designed and separating studies between situation awareness and effects of SA effectively is a sound decision.

The major concern about the study is the data analysis. I *think* the analysis of the expert vs novice is unsound. The issue is that the split of the data into two sets is done on the same data that is used to analyse the results; that is, the performance of the participants is used to split the two cohorts, but then the performance is what is used to measure effectiveness of SA on the different conditions. This is circular analysis (see https://en.wikipedia.org/wiki/Circular_analysis) and is not valid. As such, the results on the split between novice and expert are not valid and should be removed. If, on the other hand, the expert vs novice was measured in some other way (e.g. number of hours played on a particular server previously), then this split would be fine.

Some comments/questions:

- Did you validate the 20 hours Minecraft minimum? Or is this just self-rated? If you did, it would be great to explain how you did this (always a challenge for researchers!)

- The two polic-information levels in the 2nd study are interesting. Is there a reason you didn't do this in the first study? (Not a criticism, just a question).

- It would be good to get some idea of what the different surveys (mentioned at the bottom of page 7) and why they are important/useful. In particular, one page 8 (under Q2), there is a link between "teaming ability" and build speed. What is the teaming ability? Is this the measures from these surveys?

- It would be great if the results from Q2 (including subject findings) could be graphed if possible.

- How was trust measured (Q3). I assume this is perceived trust measured via a survey/questionnire? I couldn't find any details about this.

Some minor typos, etc:

- page 5: "find-tuned" --> "fine-tuned"

- page 6 and beyond: the use pf "p < 0.05" and similar -- a lot of people (including me) really like to see the p-values themselves, rather than just significant or not significant. I realise you also do "p < 0.001" (which is probably fine), but really a p=0.049 and p=0.051 are almost the same result and the arbitrary boundary of 0.05 is not so important.

- page 9: "HMT" is used for the first time and is not introduced.


**Time Spent Reviewing:**

2 hours

---

> ### Author Response · Authors · 2021-08-10
> **Reviewer jxhE Response**
>
> We thank you for your feedback regarding our studies and experiment domain. We plan to release a detailed codebase so that others can reproduce our experiment and make changes to the policy and explanation type in future work.
>
> **Related Work**
>
> We thank you for pointing us to several related works that look at xAI in human-robot teaming and intent communication. We have updated our pdf to include the citations you have mentioned.
>
> We note that the status explanation is different that the idea of communicating intent. Intent communication typically allows users to project cobot behavior into the future, allowing them to modify their own behavior accordingly to maximize performance. Study 1 (Section 4) finds that the status xAI-based does not support the projection of AI behavior, limiting the user to only perceive and comprehend the agent’s current decision-making.
>
> **Unsound Data Analysis**
>
> We did not conduct a circular analysis. The split between the novice and expert is not conducted on the same data. Our process was as follows (described in Section 5.3):
> 1.	We split our data using a performance metric on a pre-test calibration task in which the subject works without the cobot to assess/baseline performance (individual task completion times). (Section 5.2, Line 284)
> 2.	On the (separate) test task (completing the build while collaborating with the cobot), we measured performance again. (Section 3.1, Line 147)
> 3.	We sought to see whether performance on the test task differed as a function of the split on the calibration task and find significance ($F(1,26)=23.5; p<0.001$). (Section 5.3, Line 309)
>
> As such, our analysis was not circular.
>
> **Did you validate the 20 hours Minecraft minimum? Or is this just self-rated? If you did, it would be great to explain how you did this (always a challenge for researchers!)**
>
> We validate user’s proficiency in Minecraft in two stages. The first is self-rated where the users confirm that they have 20 hours of experience in Minecraft. The second is through a timed hand-crafted tutorial level in Minecraft where participants practice collecting resources, crafting, and placing objects, each a key element within the human-machine teaming task. Per our IRB Protocol, we have a dismissal criteria for the tutorial level to ensure that participants have sufficient knowledge of the game of Minecraft. If participants take longer than ten minutes within the tutorial level, it is deemed they do not have basic knowledge of Minecraft, including understanding the controls, placing and crafting objects in the game, etc. and are dismissed from the experimentation with a pro-rated compensation.

---

> > ### Comment · Reviewer_jxhE · 2021-08-17
> > **Reviewer response**
> >
> > Thanks for the clarification on the (lack of) circular analysis, which is very helpful.
> >
> > To be honest, I'm still not entirely sure though what is happening though. Line 284 does not clarify anything for me, except to mention there was a training task. Section 5.2 (Results) states: "We find a significant effect between a participant’s teaming ability and the participant’s build speed". I assume here that "build speed" refers to the pre-test task? If so, this could be made clearer; e.g. "speed in building the house in the pre-test calibration task" and then label this a "pre-test calibration task in Section 5.2

---

> > > ### Author Response · Authors · 2021-08-17
> > > **Clarification of "Build Speed"**
> > >
> > > You are correct, "build speed" refers to the pre-test calibration task in Section 5.2. We will update our submission to clarify this point and label the individual house build as the pre-test calibration task in Section 5.2, as you mentioned.
> > > Thank you!

---

> > > > ### Author Response · Authors · 2021-08-30
> > > > **Reviewer jxhE Follow Up**
> > > >
> > > > Please let us know if we can provide any more information in support of the paper's acceptance. If our rebuttal has addressed your concerns, we kindly ask the reviewer might consider please increasing the reviewer's score? Otherwise, we would be happy to continue discussing any remaining items! Thank you again for your review.

---

### Author Response · Authors · 2021-08-10
**Thank you for your reviews**

Thank you for your reviews.

We greatly appreciate your feedback. In general, we were glad to hear the problem we study is “extremely important for all interactive AI/ML applications”, and that the task used within our domain is “reasonably rich”. We appreciate that the reviewers felt the writing was well-written and clear. We have responded to each review individually and have updated our submission to reflect the changes mentioned. We again thank you for your feedback, and we hope we have provided clarification and additional insight into our paper.

---

### Decision · Program_Chairs · 2021-09-27

**Decision:**

Accept (Poster)

**Comment:**

This paper presents a study on the effect of two types of interpretable/explainable models in human-machine teaming. By varying the type of explanation (showing the model for human intention recognition used by the robot and showing the robot's action-selection model), the authors conclude that showing both models increased participants' situation awareness, but that only the human intention recognition model decreased the overall task completion time. Furthermore, for 'expert' participants, the performance degraded with.
The paper has strengths in several aspects: The problem is significant; The study is well designed; and the paper is well-written. However, there are a few reservations: It is unclear if results can generalize to more complex settings; Some important references are missing.

After a thorough discussion among reviewers, we feel the paper contains significant contributions, and the strengths outweigh the weaknesses, assuming the authors should be clear and explicit about them. In particular, please, add a more thorough literature review to alleviate the feeling that the paper is out of context and overclaiming. Further, try to explain the scope of the setting and emphasize the importance of one of the first analyses of xAI under the sequential decision-making settings.

Lastly, the issue of fitness of the work to NeurIPS raised by one of the reviewers was discussed with the SAC and Program Chairs, and the committee was encouraged to take a broad view of the scope of NeurIPS. Our goal is also to encourage a more experimental and data-driven type of AI and Machine Learning research.